# Compression Behavior of Dental Flowable Composites—Digital Image Correlation and Numerical Analysis

**DOI:** 10.3390/ma17235853

**Published:** 2024-11-28

**Authors:** Agnieszka Chojnacka-Brożek, Aneta Liber-Kneć, Sylwia Łagan

**Affiliations:** Faculty of Mechanical Engineering, Department of Applied Mechanics and Biomechanics, Tadeusz Kosciuszko Cracow University of Technology, al. Jana Pawła II 37, 31-864 Cracow, Poland; agnieszka.chojnacka-brozek@pk.edu.pl (A.C.-B.); sylwia.lagan@pk.edu.pl (S.Ł.)

**Keywords:** dental flowable composite, Poisson’s ratio, digital image correlation, finite element analysis, compression

## Abstract

In the development of restorative materials, it is important to evaluate the elastic properties of the material in order to achieve good clinical results. The aim of this study was to evaluate the compression behavior of two dental flowable materials (EverX Flow and Flow-Art) using experimental methods and numerical simulation. The Poisson’s ratio was determined using two methods of strain measurement: the electrical strain gauge method (ESG) and digital image correlation (DIC). Material constants determined in experimental studies were implemented in a numerical model, and displacement analysis was conducted using the finite element method (FEM). The tests showed higher compressive strength and modulus of elasticity for EverX Flow compared to Flow-Art. The values of the Poisson’s ratio were similar for both measurement methods, ranging from 0.27 to 0.28 for EverX Flow and from 0.30 to 0.32 for Flow-Art. This demonstrated the feasibility of the DIC method for obtaining the Poisson’s ratio values for this type of composites. Compression test conditions were reproduced in the numerical analysis. The obtained distributions of the displacement field on the surface of the sample from the DIC and numerical analyses were compared. A good match was observed between DIC displacement measurements and displacement values obtained in FEM analysis. The comprehensive approach used in the study allows us to analyze whether the results obtained in the numerical simulation correspond to the material response to the applied load and validate the model.

## 1. Introduction

The mechanical properties of dental restorative materials play an important role in matching their properties to the tooth tissue (dentin and/or enamel). A good match between the properties of the tissue and the material influences a more uniform distribution of stresses and strains at the interface, which is associated with obtaining a durable tooth restoration [1,2]. During the mastication process, teeth and restorations are subjected to complex forces (compressive, tensile, shear, and bending) that cause material deformation. Dental flowable resin composites are characterized by a lower viscosity compared to conventional resin composites. This is the result of lower filler content and/or less viscous monomers. Due to its lower viscosity, the application of the restorative material into complex cavity spaces is easier, but the lower filler content impacts the mechanical properties, making them lower in comparison to conventional hybrid composites [3,4].

A complete description of the elastic behavior of a material is based on the values of Young’s modulus, bulk modulus, shear modulus, and Poisson’s ratio. These material constants are related, so in mechanical strength analysis, for example, using finite element method (FEM), the set of the Young’s modulus and Poisson’s ratio is used to calculate the stress and strain fields in the examined material. The modulus of elasticity represents the mechanical property of a material, which can be mathematically described as the relationship between acting loads (compression or tension) and deformation in the elastic region. The difference (incompatibility) between the elastic modulus of restorations and dental hard tissue results in the possibility of failure in the interface area or marginal fracture. The Poisson’s ratio represents the ratio of the relative transverse strain to the applied load to the relative axial strain in the direction of the applied load. Several different methods (static and dynamic) have been used to determine the Young’s modulus and Poisson’s ratio of dental composites [1,5,6]. Static methods are used to measure the deformation of specimens in the axial and transverse directions during axial tensile, compressive, or flexural loading [1,6,7]. These methods require additional high-accuracy displacement monitoring systems (e.g., electrical strain gauge technique (ESG), digital image correlation method (DIC), or electronic speckle pattern interferometry method (ESPI)) [2]. In dynamic methods, ultrasound is applied to the sample, and the Poisson’s ratio and Young’s modulus are obtained from the ultrasound wave’s velocity [1,6]. The variety of test methods used, different specimen sizes, strain rates, and displacement-monitoring systems mean that results are often incomparable. The electrical strain gauge method is a contact technique of displacement measurement, while DIC and ESPI are modern contactless techniques. DIC is an optical method that uses feature tracking and recording of successive images of the surface of a deformed sample. Analysis of these images makes it possible to obtain an image of the distribution of displacement on the surface, which is used to determine the deformation field [8]. The conventional strain gauge method tends to measure the local response of the material beneath the gauge, whereas DIC measures full-field strain. It is important, especially for dental materials, to evaluate the entire displacement field to predict mechanical behavior in tooth tissue, which exhibits not only anisotropic but also heterogeneous behavior [9].

The digital image correlation technique has been applied to dental materials to assess polymerization shrinkage of dental resin [10,11,12], to measure temporal variations of material properties (Knoop hardness, elastic modulus, viscosity) during light polymerization [13], and to evaluate mechanical behavior at the interface between dental material and tooth [14]. Implementation of the DIC technique to investigate shrinkage behaviors and shrinkage stress under different light-curing protocol showed that the degree of shrinkage strain and stress decreases with an oblique irradiation direction in comparison to a vertical direction during the curing of composite restoration [15]. The same study showed that the DIC technique and numerical simulation can comprehensively analyze stress developments in restorative dentistry. The DIC technique has also been used to measure the Poisson’s ratio of different engineering materials but has been utilized rarely for dental materials. Choi et al. [16] evaluated the Poisson’s ratio for dental resin–ceramic composites under uniaxial compression using the DIC technique. The values of the Poisson’s ratio for tested specimens ranged from 0.277 to 0.306. In the same study, the elastic modulus from compression tests combined with the DIC system for resin-dental material composite was obtained and showed good agreement with values of the elastic moduli from the three-point flexural test. However, to the authors’ knowledge, the DIC technique has not been used to obtain the Poisson’s ratio for the dental flowable composites.

The purpose of this study was to analyze whether the results obtained from the numerical simulation correspond to the material’s response to the applied load. To make a comparison, it is necessary to know the crucial material constants of the dental composites under consideration. Two types of flowable dental composites were tested under compression to determine stress–strain characteristic and material constants (Young’s modulus and Poisson’s ratio). The article also addresses measurement of the Poisson’s ratio of dental flowable composites, which is relatively rare. Two independent methods of displacement measurement in the longitudinal and transverse directions to the load—electrical strain gauge method and digital image correlation—were used and compared. Both methods provide good accuracy but differ in terms of contact with the material surface, field of observation/measurement, and monitoring area. The experimentally determined material constants were used in numerical simulations using the finite element method. Displacement maps recorded in compression using the digital image correlation method were compared with the results of the finite element analysis.

## 2. Materials and Methods

### 2.1. Materials

The test specimens were produced from dental flowable composites from two manufacturers: EverX Flow–bulk shade (GC Corporation, Tokyo, Japan) and Flow-Art (Arkona, Nasutow, Poland). EverX Flow (bulk shade) is composed of matrix resin (Bis-MEPP, TEGDMA, UDMA) and 70% (*w*/*w*) of the filler (barium glass and glass fiber) and is dedicated to deep cavities (curing depth of 5.5 mm) [17]. The Flow-Art resin matrix is composed of Bis-GMA and methacrylate monomers (TEGDMA, UDMA). The filler material is barium–aluminum–silicon glass with a content of 61% (*w*/*w*). It is dedicated to deep cavities [18]. Cubic-shaped samples measuring 10 × 10 × 15 mm were prepared by casting uncured composites layer by layer (the height of a single layer was 2 mm) in polytetrafluoroethylene molds. Light-curing was performed using a LED curing unit. The samples were irritated from the top. The distal tip of the curing unit light guide was positioned directly above the specimen. The composite specimens were stored dry at 22 °C for 24 h to complete the post-cure reaction. Three samples of each type of material were produced.

### 2.2. Methods

#### 2.2.1. Compression Tests

Static compression tests were conducted using an MTS Insight 50 testing machine (MTS Insight^TM^, Eden Prairie, MN, USA) combined with a digital image correlation system (DIC). Tests were conducted under room conditions (23 °C, 65% RH) at a crosshead speed of 0.5 mm/min. The recorded force and displacement values were converted to stresses and strains. Compressive strength, failure strain, and Young’s modulus were calculated. To determine the Poisson’s ratio, two methods were additionally used to measure the displacement in the longitudinal and transverse directions to the load: digital image correlation (DIC) and an electrical strain gauge (ESG) method.

#### 2.2.2. Digital Image Correlation (DIC)

The digital image correlation technique is a noncontact optical measurement tool used to determine displacement and strain in 3D. The DIC system (Dantec Dynamics GmbH, Ulm, Germany) used in this study consisted of two 2 MPx cameras (Dantec Dynamics GmbH, Ulm, Germany) and Istra 4D V4.8.2.248 software (Dantec Dynamics GmbH, Ulm, Germany) (Figure 1). The samples were covered with a speckle pattern by painting a plane surface with white acrylic paint and then applying fine dots by spraying black acrylic paint. The paints had a matte finish to avoid unwanted glare. The image of sample surface before deformation was taken as a reference image. Then, during the compressive loading, a set of images of the sample was recorded up to a load value of 1 kN and then correlated using the software. To avoid edge effects, the field of view was limited to the region of interest (ROI), which occupies about 80% of the sample (Figure 1b). The software tracks the speckle patterns and develops a 3D mesh of the surface by using stereo-triangulation. The displacement and surface strains are then computed by comparing the newly generated surface mesh to the initial surface mesh. Based on the correlation, color maps were obtained, showing displacements (Figure 1c,d) and strains on the sample surface in two directions. Based on the measured values of strains, the Poisson’s ratio was calculated.

#### 2.2.3. Electrical Strain Gauge (ESG)

The second measurement method to obtain displacement in two directions was the electrical strain gauge (Figure 2). A strain gauge rosette KFGS-1-120-D17-16L3M2S (KYOWA Electronic Instruments Co., Ltd., Tokyo, Japan) was bonded to the surface (in the center area/point) of the sample and connected to a strain gauge bridge KYOWA PCD-300 (KYOWA Electronic Instruments Co., Ltd., Tokyo, Japan). The principal operation of the strain gauge is based on relating the change in resistance of the gauge to the strain in an object. A strain gauge is a uniaxial transducer usually made of a thin metallic wire formed into a rectangular grid. The resistance of a wire depends on its length and cross-sectional area, and the change in resistance of the metallic wire can be connected to strain by a property known as the gauge factor. Three load–unload loops (0–1 kN) were performed for all samples to record strain in the longitudinal and transverse directions to the load. Based on the measured values of strains using the electrical strain gauge method and digital image correlation, the Poisson’s ratio was calculated for each strain measurement and compared.

#### 2.2.4. Finite Element Modeling (FEM)

As a complement to the DIC experimental results represented as contour displacement distributions, the finite element modeling (FEM) of the displacement fields of the specimen was performed. The modeling of the calculation schemes was carried out using the commercial finite element software package ANSYS 2022R2 (Ansys, Inc., Canonsburg, PA, USA). The model integrates the geometry of the test specimens used in the experiment. According to the loading method of experimental test, two steel blocks were placed, respectively, at the top and the bottom of the specimen. The interactions between the specimen and plates were implemented by the classic Coulomb friction behavior. The friction coefficients considered in the simulations were 0.3. All freedoms of the bottom plate were constrained. The compressive static load (1 kN) was applied to the steel plate above the prism specimen. To define the linear elastic material response range of the material model, input material properties, including the modulus of elasticity and Poisson’s ratio, obtained from uniaxial compression experiments performed by the authors within this study were introduced. All the specimens in this study were modeled using an eight-node solid element (solid 185), and the average mesh size was 0.5 mm. Additionally, after the correlation process in the DIC system, the HDF5 (Hierarchical Data Format) file and ASCII format were used to export all data steps from Istra 4D. These data were implemented in Ansys 2022R2 software to obtain displacement maps using displacement values registered by the DIC technique. This approach allows for the comparison of results obtained using FEM (defining material properties or using displacement DIC data) and the DIC method and also indicates the difference between these methods.

#### 2.2.5. Statistical Analysis

Data reported in the present study were mean values. Significant statistical differences between groups of data were determined using the Student’s *t*-test to check whether the Poisson’s ratio obtained through two methods indicates significant differences between the mean values. The accepted level of significance was 95% (i.e., *p* ≤ 0.05). The basic assumptions of the Student’s *t*-test were checked, i.e., equality of groups (the same number of observations) and homogeneity of variance (Fisher test).

## 3. Results

### 3.1. Compression Test

The mechanical behavior during compression for the tested dental flowable composites is compared in Figure 3. The mean value of compressive strength was 175.1 ± 3.3 for Flow-Art and 241.8 ± 4.0 for EverX Flow. EverX Flow shows higher values of compressive stress (*p* = 0.001) as well as Young’s modulus of elasticity (*p* = 0.01) compared to Flow-Art (Table 1). These properties improve when the filler content in the composite is higher. EverX Flow is characterized by a higher filler content (70% (*w*/*w*)) compared to Flow-Art (61% (*w*/*w*)); in addition to the particle fillers, glass micro fibers are also present (Figure 4). This results in better performance under load. However, due to the different filler amounts and types, it is difficult to compare the relation of filler contents with the mechanical properties.

The use of the digital image correlation method during the compression test made it possible to analyze the deformation on the surface of the loaded specimen in the direction of the force and transverse to it (Figure 5). The values of longitudinal and transverse strains obtained from in-plane displacement fields were used to calculate the Poisson’s ratio of tested composites. Exemplary graphs of the longitudinal and transverse strain, as well as the dependence of the Poisson’s ratio on strain, are shown in Figure 6. The Poisson’s ratio was calculated with virtual gauge elements (in our test, it was a polygon element) marked on the object surface in the region of interest, and the diagrams of strains (longitudinal and transverse) and Poisson’s ratio were plotted. It should be emphasized that in the Scilab script used in Istra 4D V4.8.2.248 software, the mean strain over the surface of the polygon element is used to calculate the Poisson’s ratio. The strain along the *y*-axis is defined as longitudinal strain, and the strain along the *x*-axis is defined as transverse strain (Figure 6).

The average values of the Poisson’s ratio calculated from strain measurements using the DIC technique were compared with values obtained from strain gauge measurements (Table 1).

Strain measurement using the electric strain gauge method was performed when the specimen was compressed to a load of 1 kN. In order to check the repeatability of the measured deformations in the elastic range, which take on small values, the measurement was carried out in three load–unload cycles. The recorded curves are shown in Figure 7. The load–unload curves indicate a faster achievement of the intended load level of 1000 N, also indicating a higher stiffness of the EverX Flow compared to Flow-Art. Recording deformations with strain gauges made it possible to calculate the Poisson’s ratio for the tested materials. The relationship of the ratio and strain in the longitudinal direction is shown in Figure 8.

The values of the Poisson’s ratio determined using the two strain measurement methods are similar to each other (Table 1). The repeatability of the measurements for both methods was good, as shown by the low standard deviation values of 0.04 and 0.05 for DIC and 0.01 for the strain gauge method. For EverX Flow, the difference in the Poisson’s ratio values for the two methods is 0.01, while for Flow-Art it is 0.02, and it is not statistically significant different (*p* > 0.05).

### 3.2. FEM and DIC Analysis

Comparison of the results for axial specimen compression simulated using the finite element method and the experimental displacement measurement using DIC Istra4D is shown for Flow-Art in Figure 9 and Figure 10 and for EverX Flow in Figure 11 and Figure 12.

The results allow one to compare displacements as the results of axial compression of the material samples obtained from numerical simulation using the FEM analysis and measured using the DIC system. Analyzing the frontal wall section that was speckled for DIC and then masked in software Istra4D, the longitudinal displacements were in the range between −0.040 and −0.062 mm for Flow-Art material (Figure 9) and between −0.006 and −0.046 mm for EverX Flow dental resin composite (Figure 11). The transverse displacements were in the range between 0.007 and 0.016 mm for Flow-Art (Figure 10) and between −0.002 and 0.01 mm for EverX Flow (Figure 12). In the numerical simulation, the same value of compression force was applied to the specimen and the same sample section was chosen for analysis. The observed longitudinal displacement maximal values obtained in FEM simulation were −0.061 mm for Flow-Art and −0.058 mm for EverX Flow. When comparing the edge values of the displacement obtained from the digital image correlation method and the finite element method, a good agreement between the obtained values can be observed. However, the distributions of the displacement fields differ, which is due to the boundary conditions and the theoretical assumptions of the FEM model (homogeneous, isotropic material, symmetric uniaxial loading, fixation of the specimen), which is a simplified representation of the experimental conditions. Additionally, a second approach to FEM analysis was applied using the software capability of the DIC system, allowing the values of the recorded displacements to be exported to Ansys. These data were implemented into the FEM model. This allowed for a comparison of the displacement distribution obtained in the model based on material constants (Young’s modulus and Poisson’s ratio) and the model using the displacement values recorded during the experiment (DIC method). The displacement values and their distribution obtained in such FEM analysis for both analyzed directions and materials showed very good agreement with the displacement distribution obtained by the DIC method.

## 4. Discussion

During chewing, the restorative material is loaded in compression, so mechanical properties such as the Young’s modulus and Poisson’s ratio are useful in dental practice for predicting deformations that can increase the risk of microleakage, as well as for finite element numerical simulations. The present study evaluated the compression behavior of two dental flowable composites. The Poisson’s ratio was obtained using two measurements methods: the digital image correlation and electronic strain gauge method. This made it possible to assess whether the use of two different approaches to displacement measurements influenced the variations between the values of the Poisson’s ratio. The values of the Young’s modulus and Poisson’s ratio obtained in the tests were used to create a FEM model. The FEM model was made in the geometry of the test setup. This provided an assessment of the feasibility of using the DIC method to validate numerical analysis.

Due to the different test methods and sample dimensions, values of the Young’s modulus and Poisson’s ratio obtained for dentin, enamel, and some dental flowable composites are not always comparable (Table 2). Properties reported in the literature are characterized by high variability. The Poisson’s ratio of dental flowable composites is reported to range from 0.28 to 0.39 [1,19]. This study, conducted for two flowable dental composites, showed Poisson’s ratio values from 0.27 to 0.32, which is within the range of values reported in the literature. The tested composites differ in filler content, but both can be classified as “low-fill” with filler content between 50% and 74% [20]. Flow-Art, with lower filler content, has a higher Poisson’s ratio value due to its lower stiffness. Tested composites also differ in the type of filler. In addition to the particulate fillers, EverX Flow composite contains glass fibers with an average length of 140 µm and a diameter of 6 µm [17]. Thus, comparing the impact of one single variable (e.g., filler content) on a given property is difficult. There are many variables (monomer type, filler shape and content, filler size and its distribution) that affect dental composite properties [20].

A comparison of the strain gauge and digital image correlation methods used to determine the Poisson’s ratio showed that the DIC method can be successfully used for dental flowable composites. Some previous studies conducted for dental materials showed that the DIC can be used for measuring deformations under load [9,16]. The Poisson’s ratio determined under compression loading by the DIC for four dental resin composites showed a value in the range from 0.277 (polymer-infiltrated-ceramic network (86 wt.%)) to 0.306 (nanoparticle resin–ceramic (71 wt.%)) [16]. Also, comparison of the elastic modulus values determined by the DIC showed good agreement with those determined during 3-point flexural bending [16].

The determined Poisson’s ratio from the two displacement measurement methods had similar values. This demonstrated the applicability of the digital image correlation method for dental flowable composites in this type of measurement. In material testing using the strain gauge method, it is assumed that the material behaves uniformly throughout its volume. The strain results from the strain gauge measurement base (the contact surface of the strain gauge with the material) are distributed over the entire surface of the tested element. The digital image correlation method allows for determining a much wider area (with the possibility of narrowing it down) that will be subjected to strain analysis and is additionally non-contact (without attaching a gauge to the specimen), which is beneficial in many biomechanical tests.

The values of the Young’s modulus of dental tissues and materials have been obtained from several tests, such as static tensile, compression, or nano-indentation. A review of studies [23,24] in this area shows a dispersion of the elastic moduli values of enamel and dentin in the range from 40 to 150 GPa and from 11 to 20 GPa, respectively, obtained in the compression test. In another review [25], much more dispersive values of the elastic modulus were shown for enamel, ranging from 8.2 to 95.8 GPa, and for dentin from 5.3 MPa to 13.3 GPa, obtained in the compression test. For nano-indentation tests, these values for enamel and dentin were in the range of 3.2 to 90 GPa and of 2 to 100 GPa, respectively [25]. These values are much lower than the results referred to by the authors of [25] for the elastic moduli of the enamel and dentin specimens: 1338.2 ± 307.9 MPa and 1653.7 ± 277.9 MPa, respectively, obtained in compression test. Such a variation in the values reported in various studies results from the dependence of the mechanical properties of dental tissues on the age, sex, and test conditions [24]. Experimental data for flow-type dental materials are limited. The value of flexural strength was reported as 141.3 ± 14.8 Mpa, and the flexural modulus was 11.32 ± 0.33 GPa for EverX Flow Bulk [26]. In another study [27], the authors reported values of flexural modulus from 9.4 ± 0.6 GPa (for EverX Bulk) to 9.9 ± 0.2 GPa (for EverX Dentin). The Art-Flow material was the subject of research [28], where the authors presented the values of the flexural strength at the level of 58.4–97.0 MPa and diametral tensile strength at 38.6–48.6 MPa via different aging procedures. The Young’s modulus values obtained in this study are lower and difficult to compare with available literature data due to the different loading schemes used in the study.

The displacement distribution on the samples’ surface obtained with the use of the digital image correlation method was compared with the results of finite element method analysis. Two approaches were used in numerical analysis. In the first, material properties were defined based on the experimentally obtained values of the Young’s modulus and Poisson’s ratio. In the second analysis, the displacements of the tested materials recorded by the DIC method were implemented into Ansys. For the first analysis, a comparison of overall displacement values in the longitudinal and transverse directions to the applied load showed good agreement. The difference in the maximum value of EverX Flow displacement determined using DIC and FEM was 0.002 mm for the longitudinal direction and 0.005 for the transverse direction. For Flow-Art, it was 0.001 mm in the longitudinal direction and 0.009 mm in the transverse direction. A smaller matching can be observed in the distribution of displacement on the sample surface. To a large extent, this may be due to methodological differences between the methods used. When using the implementation of DIC displacement data into Ansys, very good agreement both for displacement values and distribution between FEM and DIC was observed.

The DIC method is error-prone when the specimen undergoes even slight rigid body rotations and displacements, changing camera–specimen distances during the test. In such cases, the consequent changes in magnification cause systematic errors in measurements. The rigid-body motion of the sample during the experiment is a very important and limiting aspect of DIC measurements applied to the sample surface. The speckle size also has a big influence on the accuracy of the strain field measurement using DIC. This is a very important issue from a practical point of view because before capturing the images of the undeformed and deformed specimen, a speckle pattern that guarantees exposing the strain field heterogeneity due to the underlying microstructure is required. The discrepancies between the experimental and numerical results depend not only on the errors associated with the usage of the DIC method but also on the simplifications related to the numerical model (perfect shape, assumption of homogeneous mechanical behavior of materials, etc.). Nonetheless, it has to be pointed out that despite the simplifications, the agreement between the results is good. The discrepancy between the reference FE solution and DIC analysis based on the digital representation of the real speckle pattern may be a useful indicator of the accuracy of the DIC analysis carried out on the basis of images captured during the experiment.

The DIC method and numerical analysis have been used as experimental validation tools for each other in several studies. The combination of DIC and FEM for strain analysis of implant-supported prostheses showed that the overall surface strain distributions are similar for both techniques. The distribution of compressive strains was similar; however, the FEM models predicted higher strain values than the DIC method [29]. Wang et al. [30] analyzed strain distribution of human dentin under three-point bending by combining the DIC and FEM techniques. The deformation distribution and values measured by DIC matched well with FEM results.

The analysis of the literature and conducted research show the possibility of using the DIC technique to validate numerical analysis. However, a comparison of the two approaches used shows that the result is dependent on the available input data, and both the material samples themselves and the measurement techniques will never be identical. This leads to different results. Therefore, a comprehensive approach based on modern and accurate measurement techniques and numerical simulations is necessary in this type of analysis.

## 5. Conclusions

The survey focused on comparing experimental methods that can offer the best accuracy in determining Poisson’s ratios for a given dental flowable composites. The results of the mechanical tests on flowable dental composites are limited; additionally, a large discrepancy in the results can be found in the literature review. An accurate and robust method is needed to assess strain and stress and obtain data for further analysis, e.g., using FEM. The DIC method was proposed to better evaluate strain distribution and, consequently, the Poisson’s ratio. To prove the feasibility of the DIC method for evaluating the Poisson’s ratio, the strain gauge method was used. The DIC, as non-contact and full-field method of measuring displacement, was found to be adequate for small investigated areas, such are the applied teeth-fillers. The next step of this study was displacement and strain distribution analysis using FEM. The deformation values measured by DIC matched well with FEM results, but the field distribution shows differences. This requires further analysis; however, it reveals the feasibility of the DIC technique in testing dental flowable composites and the possibility of using it as a tool for validating numerical analysis. The research carried out provides values for the Poisson’s ratio for flowable dental composites, which is rarely determined in experimental studies due to methodological difficulties. Generating adequate values of materials constants (i.e., the Poisson’s ratio) is useful for dental practice for the optimal selection of the proper (most suitable) dental flowable composite. It is also essential for numerical simulations using the finite element method, widely used in dental biomechanics. The results obtained can also be used in the design of materials with a view to their adaptation to natural tissues.

## Figures and Tables

**Figure 1 materials-17-05853-f001:**
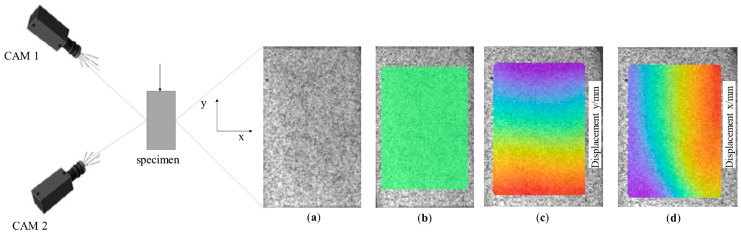
Test setup for DIC measurement: (**a**) reference image of specimen view; (**b**) deformed image of specimen with marked ROI; (**c**) the longitudinal displacement field; and (**d**) the transverse displacement field.

**Figure 2 materials-17-05853-f002:**
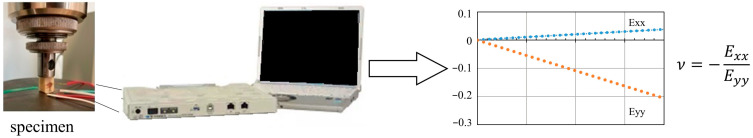
Test setup for the strain gauge measurements and calculation of the Poisson’s ratio (ν): E_xx_—strain in the transverse direction; E_yy_—strain in the longitudinal direction.

**Figure 3 materials-17-05853-f003:**
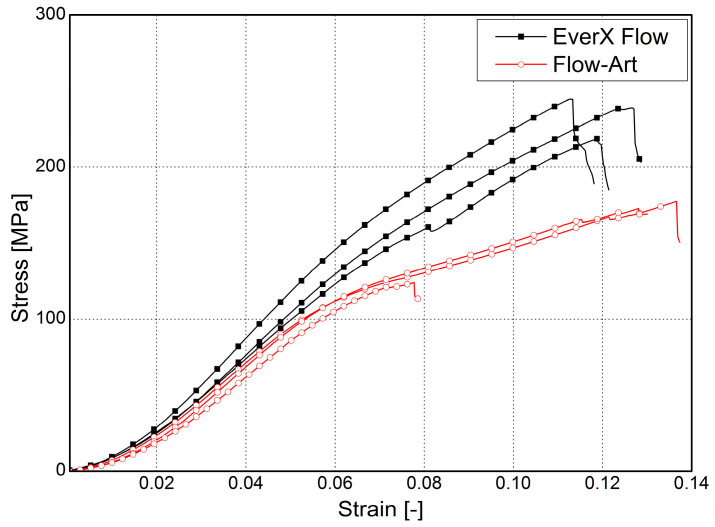
Strain–stress diagram for compression test.

**Figure 4 materials-17-05853-f004:**
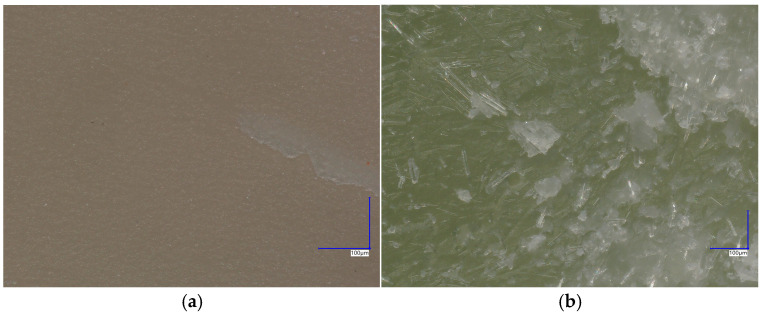
Structure of dental composite materials after compression test: (**a**) Flow-Art; (**b**) EverX Flow. The images were taken using a microscope KEYENCE VHX-7000N (Keyence International, Mechelen, Belgium).

**Figure 5 materials-17-05853-f005:**
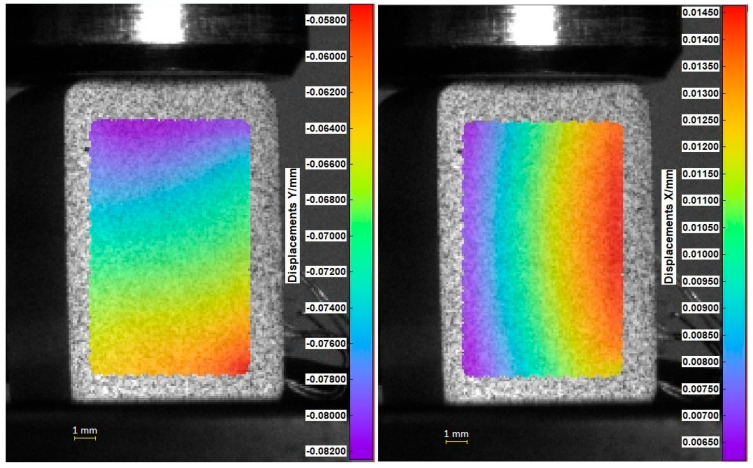
Distribution of displacement in the longitudinal (Y) and transverse (X) direction on the surface of Flow-Art for a compression force equal to 1 kN (view from Istra 4D).

**Figure 6 materials-17-05853-f006:**
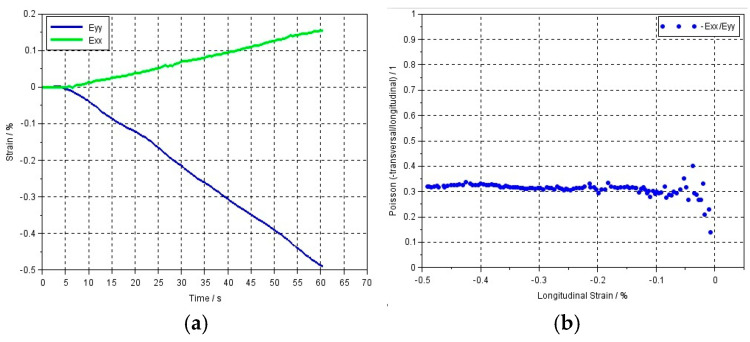
Example graphs for Flow-Art dental material from Istra4D software: (**a**) strain vs. time; (**b**) Poisson’s ratio vs. longitudinal strain.

**Figure 7 materials-17-05853-f007:**
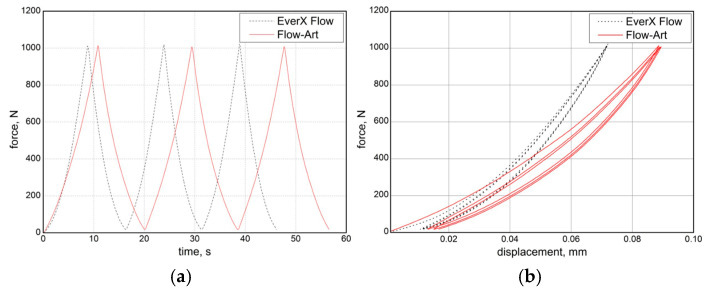
Load–unload curves: (**a**) force–time curves; (**b**) force–displacement curves.

**Figure 8 materials-17-05853-f008:**
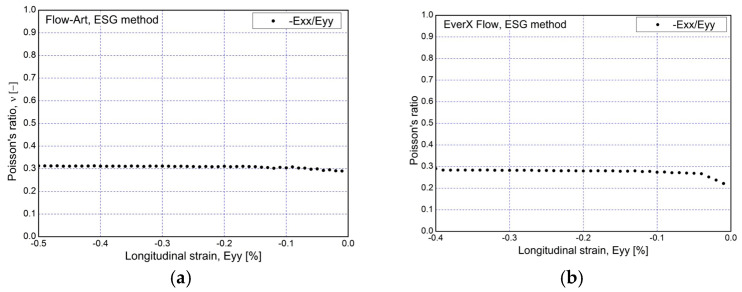
Example of Poisson’s ratio vs. longitudinal strain obtained from ESG method: (**a**) Flow-Art; (**b**) EverX Flow.

**Figure 9 materials-17-05853-f009:**
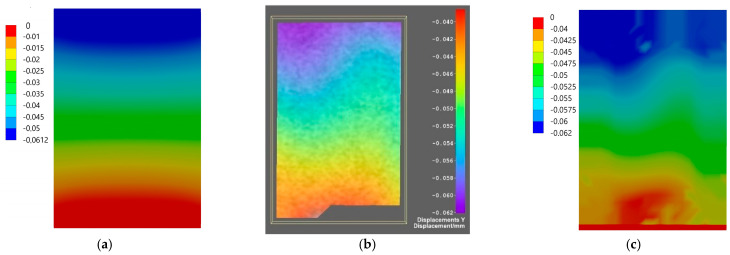
Displacement maps for axial longitudinal compression for Flow-Art dental material: (**a**) simulated using FEM; (**b**) measured using DIC; (**c**) simulated using FEM with data from DIC.

**Figure 10 materials-17-05853-f010:**
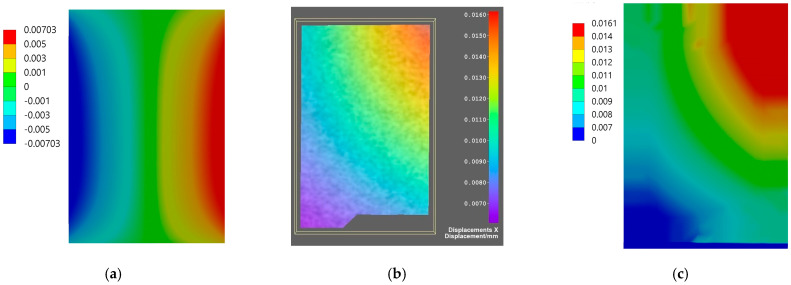
Displacement maps for transverse direction for Flow-Art dental material: (**a**) simulated using FEM; (**b**) measured using DIC; (**c**) simulated using FEM with data from DIC.

**Figure 11 materials-17-05853-f011:**
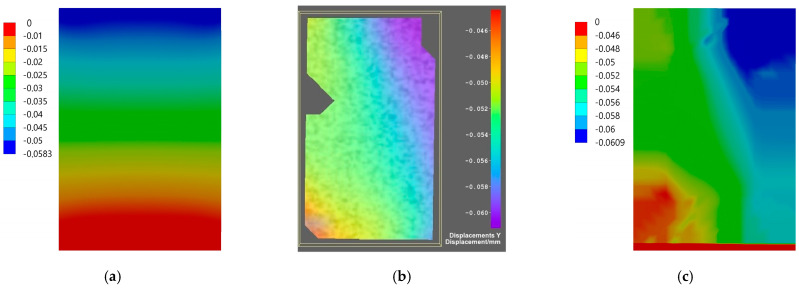
Displacement maps for longitudinal compression for EverX Flow dental material: (**a**) simulated using FEM; (**b**) measured using DIC; (**c**) simulated using FEM with data from DIC.

**Figure 12 materials-17-05853-f012:**
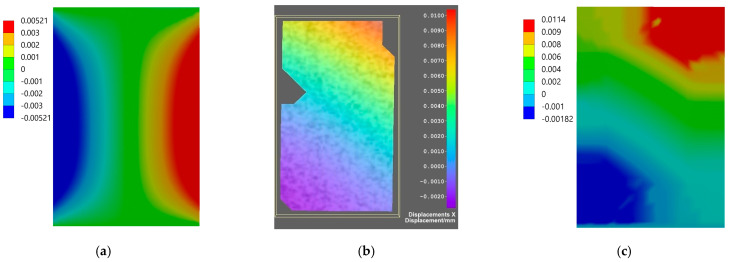
Displacement maps for transverse direction for EverX Flow dental material: (**a**) simulated using FEM; (**b**) measured using DIC; (**c**) simulated using FEM with data from DIC.

**Table 1 materials-17-05853-t001:** Comparison of material constants determined by DIC and strain gauge methods.

Material	Young’s Modulus [MPa]	Poisson’s Ratio	
		DIC	ESG
EverX Flow	2649 ± 148 ^A^	0.27 ± 0.04 ^a^	0.28 ± 0.01 ^a^
Flow-Art	2345 ± 90 ^A^	0.32 ± 0.05 ^b^	0.30 ± 0.0 1^b^

Note: Same lowercase letters within the columns indicate no statistically significant difference between the groups (*p* > 0.05). Same uppercase letters within the rows indicate statistically significant difference between the groups (*p* ≤ 0.05).

**Table 2 materials-17-05853-t002:** Poisson’s ratio of dental flowable composites0 found in the literature and present study.

Material	Manufacturer	Author	Type of Test	E [GPa]	ν [-]	
					ESG	DIC
EverX Flow	GC Corporation, Tokyo Japan	This study	Static compression	2.7 ± 0.2	0.28 ± 0.01	0.27 ± 0.04
Flow-Art	Arkona, Nasutow, Polad	This study	Static compression	2.4 ± 0.1	0.30 ± 0.01	0.32 ± 0.05
Filtek Flow	3M ESPE, Maplewood, MN, USA	[1]	Static tensile	-	0.393 ± 0.004	
Filtek Flow	3M ESPE, Maplewood, MN, USA	[21]	nano-hardness test	13.5		
Tetric Flow	Inoclar Vivadent, Schaan, Liechtenstein	[19]	nd	5.26 ± 0.05	0.28	
dentin	human	[22]	Static compression	4.04 ± 0.12	0.14 ± 0.04	
dentin	human	[23,24]	Static compression	11–20	0.3–0.31	
enamel	human			40–150	0.21–0.40	

## Data Availability

Data available within the article.

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
