# Peer review of "Compression Behavior of Dental Flowable Composites—Digital Image Correlation and Numerical Analysis"

_materials, 2024, doi:10.3390/ma17235853_

Round 1

Reviewer 1 Report

Comments and Suggestions for Authors

The manuscript describes the investigation of the compression behavior of two dental flowable materials (EverX Flow and Flow-Art) with the use of experimental methods and numerical simulations.

The topic and scope suit the journal Materials, and the special issue “Mechanical Properties of Dental Materials”; the methods are appropriate for achieving the objectives and the conclusions are supported. The limitations and advantages of the methods, future perspectives have also been discussed.

It is enlightening to see new methods being applied in dental materials science, and given the adequate rigor demonstrated within this work, it should be considered for publication in Materials, after revision.

The whole manuscript can benefit from a thorough editing. English usage needs to be improved.

Reproducibility can be improved by adding a Statistical analysis section to 2, apply analysis of the variance test such as student's t-test to test the significance of the group mean difference and reflect the results in Table 1. Would be interesting to see if the poison’s ratio obtained by the 2 methods are significantly different, or not. Sample preparation has not been introduced, the curing protocol, methods, time, storage, etc.

Some clarification may be useful: Line 161-162, “compression experiments performed by the authors” within this study? Line 250-253 is confusing. Line 293, what does “additive” refer to here?

Fig. 3, how come the black plots have data points while the red do not?

Fig. 5 b can modify the y-axis scale to eliminate the large blank space, also Fig. 7.

Table 2, enamel does not have a reference? Where did you get the values? For some materials the “Type of test” is missing.  

Please also acknowledge and cite other novel work resolving elastic properties of dental restorative materials Geometric, electronic and elastic properties of dental silver amalgam γ-(Ag3Sn), γ1-(Ag2Hg3), γ2-(Sn8Hg) phases, comparison of experiment and theory.

Comments on the Quality of English Language

Moderate editing

Author Response

Dear Reviewer,

we appreciate your thorough reviews and comments. We revised our manuscript entitled “Compression behavior of dental flowable composites – digital image correlation and numerical analysis”  according to all Reviewers’ suggestions, and marked yellow all changes in the manuscript. The manuscript has been edited and the English was also improved. Below are responses to specific comments.

Yours sincerely,

Authors

Comments 1: Reproducibility can be improved by adding a Statistical analysis section to 2, apply analysis of the variance test such as student's t-test to test the significance of the group mean difference and reflect the results in Table 1. Would be interesting to see if the poison’s ratio obtained by the 2 methods are significantly different, or not.

Response 1: Statistical analysis was performed using the Student's t-test. The section on methodology and description of results, including Table 1, was completed (marked in yellow in the manuscript).

Comments 2: Sample preparation has not been introduced, the curing protocol, methods, time, storage, etc.

Response 2: A relevant description has been added to the text.

Comments 3: Some clarification may be useful: Line 161-162, “compression experiments performed by the authors” within this study? Line 250-253 is confusing. Line 293, what does “additive” refer to here?

Response 3: Unclear information mentioned in the commentary has been corrected in the text of the paper.

Comments 4: Fig. 3, how come the black plots have data points while the red do not?

Response 4: Data points were added to the plot.

Comments 5: Fig. 5 b can modify the y-axis scale to eliminate the large blank space, also Fig. 7.

Response 5: The scale of the y-axis in Figure 5 results from the Istra software graphics and is uneditable. Therefore, the same scale is used in Figure 7. It is also used to present the value of the Poisson’s ratio in its theoretical description, i.e. in the range from 0 to 1.

Comments 6: Table 2, enamel does not have a reference? Where did you get the values? For some materials the “Type of test” is missing.  

Response 6: Missing information was completed.

Comments 7: Please also acknowledge and cite other novel work resolving elastic properties of dental restorative materials Geometric, electronic and elastic properties of dental silver amalgam γ-(Ag3Sn), γ1-(Ag2Hg3), γ2-(Sn8Hg) phases, comparison of experiment and theory.

Response 7: Due to the limited use of amalgam in modern restorative dentistry, we believe that a comparison of the composites examined in this study, e.g. due to differences in their application and their use according to cavity class, would not increase the scientific value of the study.

Reviewer 2 Report

Comments and Suggestions for Authors

Review for

materials-3317213

Compression behavior of dental flowable composites – digital image correlation and numerical analysis

This paper focuses on the compressive behavior of dental flowable composites and conducts research by combining experimental methods (digital image correlation method and electronic strain gauge method) with numerical analysis (finite element method). The overall logic is relatively clear and the structure is complete. The research topic has certain clinical significance because the mechanical properties of dental restorative materials have an important impact on the restorative effect. However, there is still room for improvement in some aspects of the paper. I think, nonetheless, that the manuscript could be improved if the authors could address the comments and recommendations I listed below.

  1. In the study of the mechanical properties of dental materials, although different measurement methods have been adopted, the overall innovation is somewhat lacking. Some research ideas and methods have been similarly applied in other material studies. It is necessary to highlight more the unique innovation points in the study of dental flowable composites. Therefore, the novelty of this research should be highlighted.
  2. Please add a scale bar in Figure 4.
  3. In the results section, the explanation of some data can be more in-depth. For example, when comparing the mechanical properties of two materials (EverX Flow and Flow - Art), in addition to mentioning the influence of filler content, further analysis can be carried out from the perspective of material microstructure, etc.
  4. The conclusion part is rather brief and could further look forward to the potential application value of the research results in dental clinical practice or material research and development.

Author Response

Dear Reviewer,

we appreciate your thorough reviews and comments. We revised our manuscript entitled “Compression behavior of dental flowable composites – digital image correlation and numerical analysis”  according to all Reviewers’ suggestions, and marked yellow all changes in the manuscript. The manuscript has been edited and the English was also improved. Below are responses to specific comments.

Yours sincerely,

Authors

Comments 1: In the study of the mechanical properties of dental materials, although different measurement methods have been adopted, the overall innovation is somewhat lacking. Some research ideas and methods have been similarly applied in other material studies. It is necessary to highlight more the unique innovation points in the study of dental flowable composites. Therefore, the novelty of this research should be highlighted.

Response 1: Some changes have been made in the purpose of the work as well as in the conclusions to emphasize the novelty of the work.

Comments 2: Please add a scale bar in Figure 4.

Response 2: The scale bar was added.

Comments 3: In the results section, the explanation of some data can be more in-depth. For example, when comparing the mechanical properties of two materials (EverX Flow and Flow - Art), in addition to mentioning the influence of filler content, further analysis can be carried out from the perspective of material microstructure, etc.

Response 3: The results and discussion section has been supplemented with a commentary on the composition of the test materials and their influence on the results obtained (highlighted in yellow in the manuscript). We also added Figure 4 to show differences in microscopic structure of tested materials.

Comments 4: The conclusion part is rather brief and could further look forward to the potential application value of the research results in dental clinical practice or material research and development.

Response 4: The conclusions section was supplemented by the application potential of the results obtained (highlighted in yellow in the manuscript).

Reviewer 3 Report

Comments and Suggestions for Authors The manuscript by Chojnacka-Brożek A. et al, entitled: Compression behavior of dental flowable composites – digital image correlation and numerical analysis, presents an interesting and valuable study on the mechanical properties of two dental composites used for deep cavity filling. The manuscript is well written and well structured, and the results are well presented. However, the English language needs to be revised and the authors should avoid repetition. Also, all abbreviations need to be explained in the text of the manuscript (some are only mentioned in the abstract). The methods are well described but need to be structured more logically (currently they are mixed and written like a story). The research was well conducted and organized and provided valuable insights into the compressibility and flexibility of the materials. Considering that the manuscript is more of a case study on only two commercially available materials and few analyzes were conducted, the discussion section becomes the most important section of the manuscript. The authors discuss the results in an appropriate manner and the figures are relevant to the results of the study. The results are well presented, the discussions are detailed, and the data are correctly interpreted. Nevertheless, some discussion of the correlation between mechanical performance and composition of materials is needed. Please emphasize the role of each ingredient on the compressive performance of the samples. Comments on the Quality of English Language

English language requires minor revision

Author Response

Dear Reviewer,

we appreciate your thorough reviews and comments. We revised our manuscript entitled “Compression behavior of dental flowable composites – digital image correlation and numerical analysis”  according to all Reviewers’ suggestions, and marked yellow all changes in the manuscript.

The manuscript has been edited and the English was also improved.

Below are responses to specific comments.

Yours sincerely,

Authors

Comments 1: Also, all abbreviations need to be explained in the text of the manuscript (some are only mentioned in the abstract).

Response 1: The abbreviations in the introduction section are explained.

Comments 2: The methods are well described but need to be structured more logically (currently they are mixed and written like a story).

Response 2: The structure of the Materials and Methods section has been structured, the individual research methods are described in the subsections.

Comments 3: Nevertheless, some discussion of the correlation between mechanical performance and composition of materials is needed. Please emphasize the role of each ingredient on the compressive performance of the samples.

Response 3: The results and discussion section has been supplemented with a commentary on the composition of the test materials and their influence on the results obtained (highlighted in yellow in the manuscript). We also added Figure 4 to show differences in microscopic structure of tested materials.

Round 2

Reviewer 2 Report

Comments and Suggestions for Authors

well improved

Reviewer 3 Report

Comments and Suggestions for Authors

Thank you for the replies